# ANYECG: FOUNDATIONAL MODELS FOR ELECTRO-CARDIOGRAM ANALYSIS

## ABSTRACT

Electrocardiogram (ECG), a non-invasive and affordable tool for cardiac monitoring, is highly sensitive in detecting acute heart attacks. However, due to the lengthy nature of ECG recordings, numerous machine learning methods have been developed for automated heart disease detection to reduce human workload. Despite these efforts, performance remains suboptimal. A key obstacle is the inherent complexity of ECG data, which includes heterogeneity (*e.g.*, varying sampling rates), high levels of noise, demographic-related pattern shifts, and intricate rhythm-event associations. To overcome these challenges, this paper introduces AnyECG, a foundational model designed to extract robust representations from any real-world ECG data. Specifically, a tailored ECG Tokenizer encodes each fixed-duration ECG fragment into a token and, guided by proxy tasks, converts noisy, continuous ECG features into discrete, compact, and clinically meaningful local `rhythm codes`. These codes encapsulate basic morphological, frequency, and demographic information (*e.g.*, sex), effectively mitigating signal noise. We further pre-train the AnyECG to learn rhythmic pattern associations across ECG tokens, enabling the capture of cardiac event semantics. By being jointly pre-trained on diverse ECG data sources, AnyECG is capable of generalizing across a wide range of downstream tasks where ECG signals are recorded from various devices and scenarios. Experimental results in anomaly detection, arrhythmia detection, corrupted lead generation, and ultra-long ECG signal analysis demonstrate that AnyECG learns common ECG knowledge from data and significantly outperforms cutting-edge methods in each respective task.

## 1 INTRODUCTION

The electrocardiogram (ECG) is a widely used test that records the heart's electrical activity, facilitating the monitoring and diagnosis of various cardiac conditions. Due to variations in ECG devices, recording conditions, patient characteristics, the length of recorded ECG signals, the number of leads, the sampling rates, as well as the signal-to-noise ratio (SNR), can vary significantly. For example, in non-clinical settings, wearable devices typically collect long-term single-lead or dual-lead ECG signals at lower sampling rates, covering a variety of human activity scenarios, which often results in higher noise levels (Abbaspourazad et al., 2023; Ansari et al., 2023). In contrast, standard devices used in hospital outpatient clinics capture high-resolution eight-, twelve-, or eighteen-lead ECG signals in a resting state for diagnostic purposes (Herman et al., 2024). Additionally, the noise in ECG data can originate from device artifacts, baseline wander, muscle noise, as well as external interference (Singh & Sharma, 2022). These heterogeneity and complexity present major challenges in developing a unified model that can effectively handle ECG signals recorded across various devices, scenarios, and clinical purposes.

Sequence models, such as large language models, developed using large-scale data in the wild have shown significant advantages in learning robust representations and demonstrated robustness in downstream tasks. However, adapting sequence model pre-training approaches to ECG data in real-world settings poses unique challenges: **(1) Heterogeneity:** real-world ECG signals vary in length, sampling rate, and the number of channels due to differences in devices and scenarios. A unified model (*e.g.*, with fixed tokenizer settings) is needed to effectively manage this diversity while maintaining temporal resolution and avoiding the introduction of artifacts. **(2) Low SNR:** ECG signals inherently have a low SNR, and pathological waveforms are often subtle, making it easy for noise

to interfere with the understanding of critical features. **(3) Demographic shift:** ECG waveforms can vary due to patient demographics (*e.g.*, age, sex, ethnicity). For instance, pediatric ECGs differ from adults in disease presentation and heart rate (Chen et al., 2024), and distinct ethnic groups may exhibit unique ECG characteristics (Jain et al., 2010), which hinder models from generalizing across diverse populations. **(4) Implicit rhythm-event association:** The systematic arrangement of rhythm patterns may indicate some cardiac events. ECG analysis depends on identifying these rhythm associations and event patterns. However, noisy real-world ECG data make it difficult for models to capture sequential relationships, hindering the accurate cardiac event detection.

To overcome these challenges, we introduce AnyECGs, a family of ECG foundational models designed for robust representation learning on ECG signals in diverse, real-world settings. The development of AnyECG involves two main pre-training phases: the ECG Tokenizer pre-training and the entire AnyECG foundation model pre-training. The first phase captures key local rhythmic patterns from noisy ECG signals, while the second learns associations across the rhythmic patterns that implies cardiac events. Specifically, we first pad any ECG signals to unified length, segment a signal into a collection of fixed-duration fragments, project each fragment into a token orderly, and pad missing channels, standardizing the diversity in sampling rates, lengths, and channel numbers. We further design a new hierarchical modeling approach to tackle ultra-long ECG signal (solving challenge **(1)**). In the ECG Tokenizer pre-training phase, a `Rhythm Codebook` is established to capture the key local morphological and frequency features inherent in ECG signals. The ECG Tokenizer extracts ECG features that are closely aligned with these `Rhythm Codes`, effectively reducing noise by matching the input patches to those representative codes (solving challenge **(2)**). Additionally, the extracted ECG features are also required to recover demographic information about the patient (addressing challenge **(3)**). Then, we apply 'masked modeling' approach in the AnyECG pretraining phase, where the model predicts `Rhythm Code` indices to fill in masked patches. This approach encourages the recovery of masked ECG patches based on their relationship with unmasked ECG patches, enabling the model to learn cardiac event semantics that are essential for downstream tasks, thus addressing challenge **(4)**. With these designs, AnyECG can facilitate knowledge transfer across various ECG sources in the wild, enabling to learn shared ECG and cardiac event knowledge that is applicable to downstream tasks. Our contributions are listed below.

- **ECG Foundational Model:** We introduce AnyECG, a foundational ECG model that unifies representation learning by capturing important local rhythm patterns in ECG signals and their semantic relationships, providing a flexible framework adaptable to any ECG signals for various downstream applications.
- **ECG Tokenizer:** We present an ECG tokenizer that extracts compact, noise-resilient `Rhythm Codes` utilizing the Multi-View Synergistic Decoder that reconstructs these codes from Morphology, Frequency, and Demography perspectives to capture essential diagnostic features and improve generalization across diverse populations.
- **Various Downstream Tasks Adaptability:** By pretrained to learn the associations among ECG tokens, our AnyECG is sensitive to potential cardiac events, demonstrating strong generalization capabilities across various downstream tasks, including anomaly detection, arrhythmia detection, corrupted lead generation, and ultra-long ECG signal analysis.

## 2 METHODOLOGY

Our proposed AnyECG adopts a common Transformer architecture, incorporating a novel attention module and a special tokenizer that can be adapted to both the self-supervised learning pretraining pipeline and various downstream tasks for any ECG signals, as shown in Figure 1. The self-supervised learning process of AnyECG is divided into two phases: pretraining the ECG Tokenizer and pretraining the AnyECG backbone.

### 2.1 ANYECG ARCHITECTURE

**ECG Signal Pre-Processing.** To preserve the natural characteristics of the ECG signals, we applied minimal pre-processing steps aimed at reducing noise while maintaining signal integrity. First, we used a bandpass filter between 0.1 Hz and 75 Hz to remove low-frequency noise, followed by a notch filter at 50 Hz to eliminate power-line interference. The ECG signals were then resampled to

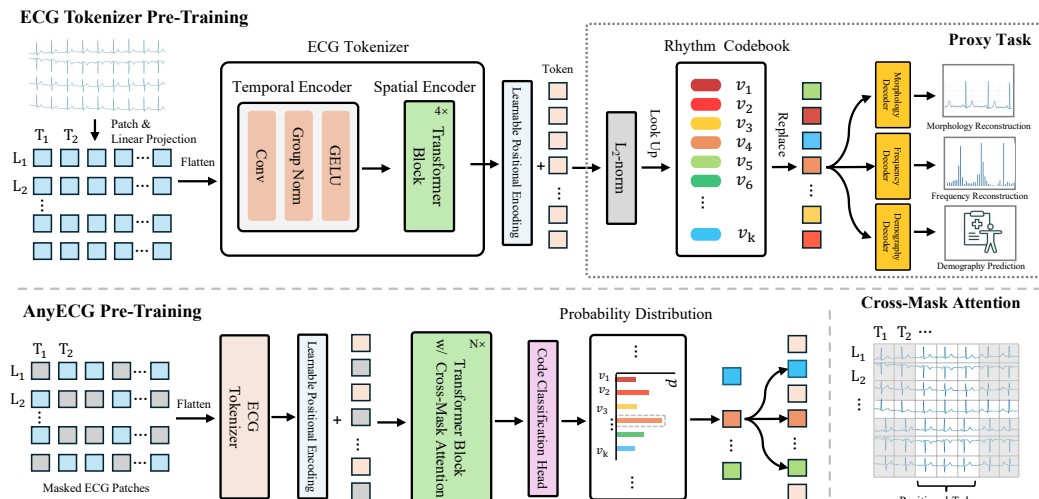

Figure 1: **Overall architecture and pre-training pipeline of AnyECG.** AnyECG is pre-trained in two steps. The Patient Attribute Tokenizer is pre-trained through proxy tasks to embed morphology, frequency, and demography into tokens **(up)**. Then, the entire AnyECG, along with the Patient Attribute Tokenizer, is further pre-trained by predicting the code indices of the masked tokens to learn the semantic relationships between tokens **(bottom-left)**. The Cross-Masking approach restricts interactions of patches from the same lead or from the same position across different leads **(bottom-right)**. LN: LayerNorm, Conv: 1D convolution with kernel size of 15.

300 Hz to standardize the sampling rate across all data sources, as 300 Hz is considered sufficient for diagnosing most cardiac conditions based on the Nyquist-Shannon sampling theorem. Finally, wavelet-based denoising was performed using the 'db6' wavelet, following (Ma et al., 2022). To align with the Transformer input format, a multi-channel ECG signal $X \in \mathbb{R}^{L \times T}$ (where $L$ represents the number of ECG leads (channels), and $T$ denotes the total number of recorded temporal points) was segmented along the time axis into fixed-duration patches of size $w$. This divides each lead into $P$ patches, where $P$ is the minimal positive integer satisfying $P \times w \geq T$. If $T$ is not divisible by $s$, we pad the signal with zeros at the end to reach a length of $Ps$. Each patch $x_{j,k} \in \mathbb{R}^s$ is defined as $x_{j,k} = X_{j,(k-1)s+1:ks}$, where $j = 1, 2, \ldots, L$ denotes the lead index and $k = 1, 2, \ldots, P$ denotes the patch index along the time axis. The total number of patches is $N = L \times P$, and we flatten them into a patch sequence $X' \in \mathbb{R}^{N \times w}$ of length $N$ before feeding it into the model.

**ECG Tokenizer.** The objective of ECG Tokenizer is to effectively capture both the temporal and spatial features of ECG signals and generate an embedding $H \in \mathbb{R}^{N \times d}$ from $X'$. We firstly use a temporal encoder to learn local temporal patterns by independently processing each ECG patch $x_{j,k}$. The temporal encoder consists of one 1D convolution, a group normalization, and a GELU activation function. Then, a spatial encoder with 4 Transformer blocks is used. Each patch $x_{j,k}$ is processed by the temporal encoder and spatial encoder sequentially to obtain an embedding $h'_{j,k} \in \mathbb{R}^d$. To enhance the model's understanding of the temporal sequence and leads relationships, learnable temporal position encoding $\tau_k \in \mathbb{R}^d$ and lead position encoding $\sigma_j \in \mathbb{R}^d$ are along the temporal dimension $k$ and channel dimension $j$, separately. The output of the ECG Tokenizer is the temporal-spatial encoding $h_{j,k} = h'_{j,k} + \tau_k + \sigma_j$.

**Cross-Mask Attention (CMA).** Unlike other sequential data like text, ECG signals typically include multiple leads, with the signals at the same positions across leads providing complementary information (Chen et al., 2021). Therefore, in contrast to conventional multi-head self-attention, we introduce CMA, which differentiates the structure of our AnyECG. CMA allows each patch to interact only with patches within relevant channels (*i.e.* leads) and temporal contexts. We apply CMA as the attention module within the Transformer blocks of both the ECG Tokenizer and AnyECG backbone. Given input $H$, $Q = \text{LayerNorm}(H)W_Q$, $K = \text{LayerNorm}(H)W_K$, $V = HW_V$, where

$W_Q, W_K, W_V \in \mathbb{R}^{d \times d_{\text{model}}}$ are learnable projection matrices, LayerNorm($\cdot$) denotes layer normalization, and $d_{\text{model}}$ is the model dimension. The CMA is computed as:

$$\text{CMA}(Q, K, V) = \text{softmax}\left(\frac{QK^\top + M}{\sqrt{d_{\text{head}}}}\right)V, \quad M_{i,j} = \begin{cases} 0, & \text{if } j \in \mathcal{A}(i) \\ -\infty, & \text{otherwise} \end{cases} \quad (1)$$

where, $d_{head}$ is the number of attention head; $M \in \mathbb{R}^{N \times N}$ is the attention mask matrix. $\mathcal{A}(i)$ includes patches from the same lead $j$ or the same position, as illustrated in Figure 1 bottom right. Notably, a positional tolerance (mask width) is used to improve the model's robustness, accounting for slight delays in certain leads caused by variations in cardiac signal conduction, which is particularly significant for some diseases. In AnyECG, we adopt the multi-head attention version.

## 2.2 ECG TOKENIZER PRETRAINING

### 2.2.1 ECG TOKENIZER WITH RHYTHM CODEBOOK

**Motivation.** ECG signals are inherently high-dimensional time-series data, often characterized by a low SNR due to sparse key information and contamination from various types of noise. To address these issues, we propose a vector-quantized rhythm codebook that transforms raw ECG signals into compact, discrete tokens, enabling robust and noise-resistant representation learning. The transformation of rhythm codebook enhances low-SNR signals into a high-SNR representation, accurately capturing true cardiac activity while minimizing the effects of noise.

Initially, each patch $x_{j,k} \in \mathbb{R}^s$ represents a portion of the signal over $w$ time steps in lead $j$. The tokenizer processes these patches into feature representations, yielding embeddings $h_{j,k} \in \mathbb{R}^d$, where $d$ is the dimension of each embedding. To discretize these continuous embeddings into tokens suitable for subsequent processing, we employ a quantizer that maps each embedding $h_{j,k}$ to the nearest codeword in a predefined codebook $V$. The codebook $V \in \mathbb{R}^{K \times d}$ consists of $K$ codes $v_1, v_2, \ldots, v_K$. The quantization process is defined as:

$$i^* = \underset{i \in \{1,2,\ldots,K\}}{\arg\min} \left\| \frac{h_{j,k}}{\|h_{j,k}\|_2} - \frac{v_i}{\|v_i\|_2} \right\|^2 \quad (2)$$

where $\| \cdot \|_2$ denotes the $\ell_2$-norm, and normalization ensures that the distance measure is equivalent to maximizing the cosine similarity. The assigned discrete token index for the patch $x_{j,k}$ is index $i^*$. This process effectively quantizes the ECG signal into a sequence of discrete tokens $\{z_{j,k}\}$, reducing the influence of noise and enhancing the signal quality.

By transforming the ECG data into a low dimension and high-SNR tokenized representations $z_{j,k}$, the ECG Tokenizer enables the model to focus on the meaningful aspects of the cardiac signal, such as heartbeat patterns and rhythms, which improves the model's ability to generalize across different datasets.

### 2.2.2 MULTI-VIEW SYNERGISTIC DECODER

To better capture the demographic variations and morphological changes inherent in ECG signals, we propose a **Multi-View Synergistic Decoder** containing three decoders for different proxy tasks.

**Morphology Decoder** aims to reconstruct the original temporal ECG signals, focusing on preserving time-domain information critical for identifying features like QRS complexes and arrhythmia. By reconstructing the time-domain signals, we ensure that the essential temporal characteristics of the cardiac cycles are retained, providing a foundation for accurate heartbeat analysis. The reconstruction loss for the Morphology Decoder is defined as:

$$\mathcal{L}_{\text{morphology}} = \sum_{j=1}^{L} \sum_{k=1}^{P} \left\| o_{j,k}^m - x_{j,k} \right\|_2^2 \quad (3)$$

where $o_{j,k}^m$ is the output of the Morphology Decoder for patch $x_{j,k}$.

**Frequency Decoder** predicts the frequency characteristics of ECG signals by incorporating frequency-domain information, which is essential for capturing periodic and spectral features associated with cardiac conditions. Unlike traditional methods that focus solely on time-domain or frequency features, this decoder leverages the Discrete Wavelet Transform (DWT)(Shensa et al., 1992) to analyze the signals simultaneously in both time and frequency domains. For each ECG patch $x_{j,k} \in \mathbb{R}^s$, corresponding to lead $j$ and patch index $k$, we apply the DWT to decompose the time-domain signal into wavelet coefficients, capturing localized frequency content. The DWT performs a multi-scale decomposition of the signal recursively, obtaining features across different frequency ranges. The wavelet decomposition process consists of two main parts. At the initial stage, the original signal is the approximation coefficients at level zero, $c_A^{(0)} = x_{j,k}$. Then, at the Recursive Decomposition stage, for each level $l$ ($l = 1, 2, \ldots, L_w$), we use the approximation coefficients from the previous level $c_A^{(l-1)}$ to obtain the current level's approximation coefficients $c_A^{(l)}$ and detail coefficients $c_D^{(l)}$ through convolution and downsampling:

$$c_A^{(l)}[n] = \sum_m c_A^{(l-1)}[m] \cdot g[2n - m] \qquad c_D^{(l)}[n] = \sum_m c_A^{(l-1)}[m] \cdot h[2n - m] \qquad (4)$$

where $g[\cdot]$ and $h[\cdot]$ are the coefficients of the low-pass and high-pass filters, respectively, $n$ is the index of the downsampled coefficients, the convolution operation captures the signal's features in the corresponding frequency range, and downsampling reduces the resolution, focusing on lower-frequency components. We obtain a hybrid time-frequency representation of the ECG signal through multi-scale decomposition by performing these operations on the approximation coefficients $c_A^{(l-1)}$ at each level, simultaneously capturing both the low-frequency (approximation coefficients) and high-frequency (detail coefficients) information of the signal. For stable convergence during training, we apply z-score normalization to the frequency magnitudes within each patch. The reconstruction loss for the Frequency Decoder is defined as:

$$\mathcal{L}_{\text{freq}} = \sum_{l=1}^{L_w} \left( \left\| \hat{c}_A^{(l)} - c_A^{(l)\,\text{norm}} \right\|_2^2 + \left\| \hat{c}_D^{(l)} - c_D^{(l)\,\text{norm}} \right\|_2^2 \right) \qquad (5)$$

where $\hat{c}_A^{(l)}$ and $\hat{c}_D^{(l)}$ are the predicted approximation and detail coefficients at level $l$, respectively, and $c_A^{(l)\,\text{norm}}$ and $c_D^{(l)\,\text{norm}}$ are the corresponding normalized actual coefficients. The loss is computed across all decomposition levels $l$ from 1 to $L_w$.

**Demography Decoder** predicts patient-specific attributes (e.g., age, weight, or other demographic factors), represented as a vector $a \in \mathbb{R}^{d_a}$. By jointly predicting these attributes, the model gains a personalized understanding of the patient's condition. This personalized aspect allows the model to better account for inter-patient variability, which is critical in making accurate clinical predictions. The loss for the Demography Decoder is defined as:

$$\mathcal{L}_{\text{demography}} = \|o^a - a\|_2^2 \qquad (6)$$

where $o^a$ represents the predicted patient-specific attributes, and $a$ is the ground truth patient attribute vector.

**Overall Loss Function for ECG Tokenizer** In addition to reconstruction loss functions from all decoders, we also include codebook loss and commitment loss to ensure that the quantized tokens remain faithful to the original signal and stabilize the training process. The codebook loss and commitment loss are defined as:

$$\mathcal{L}_{\text{codebook}} = \sum_{j=1}^{C} \sum_{k=1}^{P} \left\| \text{sg}\left(h_{j,k}\right) - v_{z_{j,k}} \right\|_2^2 \quad \mathcal{L}_{\text{commitment}} = \beta \sum_{j=1}^{C} \sum_{k=1}^{P} \left\| h_{j,k} - \text{sg}\left(v_{z_{j,k}}\right) \right\|_2^2 \qquad (7)$$

where $h_{j,k}$ is the embedding of the patch $x_{j,k}$, $v_{z_{j,k}}$ is the codebook vector corresponding to the token $z_{j,k}$, $\text{sg}(\cdot)$ denotes the stop-gradient operator, and $\beta$ is a weighting coefficient for the commitment loss. The overall loss function combines all components:

$$\mathcal{L}_T = \mathcal{L}_{\text{morphology}} + \mathcal{L}_{\text{frequency}} + \mathcal{L}_{\text{demography}} + \mathcal{L}_{\text{codebook}} + \mathcal{L}_{\text{commitment}} \tag{8}$$

This loss function requires the reconstruction of both the temporal and frequency components of the ECG signal, while also ensuring the recovery of patient-specific factors for personalized modeling. Experiments in Appendix 7.4 shows the importance of each component in the total loss function.

## 2.3 ANYECG MASKED PRE-TRAINING

Inspired by self-supervised learning from masked modeling in NLP (Kenton & Toutanova, 2019) and vision (Bao et al., 2021; He et al., 2022), we design a hybrid-scale masked ECG modeling strategy, where random segments of ECG signals are masked and the model is learned to reconstruct missing parts.

After using ECG Tokenizer process $X'$ to embeddings $H \in \mathbb{R}^{N \times d}$, we randomly generate a mask $M \in \mathbb{R}^{N \times 1}$, where its component $m_{j,k} \in \{0, 1\}$. The masked patches are replaced with a learnable mask token $h_M \in \mathbb{R}^d$. The masked embeddings $\tilde{h}_{j,k}$ are defined as: $\tilde{h}_{j,k} = (1 - m_{j,k}) \cdot h_{j,k} + m_{j,k} \cdot h_M$. These augmented embeddings $\tilde{h}_{j,k}$ are then reshaped into a sequential format and fed into a Transformer encoder to generate contextualized representations $\tilde{h}'_{j,k} \in \mathbb{R}^d$. Each contextualized vector $\tilde{h}'_{j,k}$ is passed through a linear classifier followed by a softmax function to produce a probability distribution over the codebook tokens $V = \{v_1, v_2, \ldots, v_K\}$: $p(v_i \mid \tilde{H}) = \text{softmax}\left(W\bar{h}'_{j,k} + b\right)_i$, where $\bar{H}$ denotes the collection of all augmented embeddings $\tilde{h}_{j,k}$, and the subscript $i$ refers to the $i$-th element of the output vector. The training objective for the masked modeling process is to minimize the negative log-likelihood of predicting the correct tokens $v_{z_{j,k}}$ at the masked positions:

$$\mathcal{L}_{\text{mask}} = -\sum_{j=1}^{L}\sum_{k=1}^{P} m_{j,k} \cdot \log p\left(v_{z_{j,k}} \mid \tilde{H}\right) \tag{9}$$

The masked pretraining facilitates the model in learning generic representations from the input data by capturing the implicit rhythm-event associations and sequential relationships crucial for ECG analysis, thereby enhancing its ability to capture the underlying cardiac event patterns in the ECG signals.

## 3 DOWNSTREAM APPLICATION

This section evaluates AnyECG's performance across multiple ECG datasets to prove its generality. In Section 3.1, we summarize datasets utilized in the experiments. Section 3.2 explains the experimental setup in detail. In Section 3.3, we present the results of our experiments, benchmarking AnyECG against state-of-the-art methods across multiple tasks, including anomaly detection, arrhythmia detection, ECG lead generation, and ultra-long ECG sequence recognition. In Section 7.4 and 7.3, we also present **ablation studies** on hyperparameter selection and the necessity of two-stage pre-training.

## 3.1 ECG DATASETS

To evaluate the performance of AnyECG and baseline models, we utilized a comprehensive set of ECG datasets that include all available unlabeled data during pretraining. These datasets cover a wide spectrum of cardiac conditions, patient demographics, and recording scenarios, ensuring robust testing across diverse settings. For various downstream tasks, we mixed all datasets together to minimize biases introduced by individual datasets and to better validate the model's generalizability. This approach reduces the discrepancies arising from different data sources and enhances the unified

Table 1: Summary of ECG Datasets

| Dataset | Recordings | Sampling Rate | Duration | Notes |
|---------|-----------|---------------|----------|-------|
| CPSC (Liu et al., 2018) | 6877 | 500 Hz | 6–60 s | Balanced sex |
| CPSC-Extra (Liu et al., 2018) | 3453 | 500 Hz | 6–60 s | Balanced sex |
| INCART (Tihonenko et al., 2008) | 74 | 257 Hz | 30 min each | High-res; arrhythmia |
| PTB (Bousseljot et al., 1995) | 516 | 1000 Hz | Varies | Wide range of pathologies |
| PTB-XL (Wagner et al., 2020) | 21837 | 500 Hz | 10 s | Extensive clinical ECGs |
| G12EC | 10344 | 500 Hz | Varies | The Southeast's unique demographics |
| Undisclosed Dataset | 10000 | 500 Hz | 6–60 s | Geographically distinct test set |

capability of the model. The detailed data construction of the datasets can be found in Table 1. All datasets are formatted in WFDB format, including associated binary and text files that detail signal attributes and clinical annotations using SNOMED-CT codes. Detailed information about the datasets is provided in Appendix 7.1.

## 3.2 EXPERIMENTAL SETUP

**Model Configurations.** We introduce three configurations of AnyECG: AnyECG-B, AnyECG-L, and AnyECG-XL, containing 254M, 500M, and 1.7B parameters, respectively. The increase in parameters is achieved by deepening the Transformer encoder and expanding the hidden layer sizes. To maintain consistency across all configurations, we set the patch size $P = 300$, which corresponds to 1 second of ECG data. The maximum sequence length is fixed at 1,024 tokens, sufficient for most ECG applications. During ECG Tokenizer training and AnyECG pre-training, sequences shorter than this length are padded. To preserve data integrity, we mask the attention values associated with these padding tokens.

**Training Environment.** The pre-training of AnyECG was conducted on a comprehensive dataset compiled from seven different sources. For the downstream tasks, data splitting followed standard procedures, dividing the data into training and validation subsets using an 80/20 ratio. Binary cross-entropy loss was employed for binary classification tasks, while cross-entropy loss was utilized for multi-class classification tasks. Evaluation metrics for the downstream tasks are detailed in the Appendix 7.2. All experiments were executed on a computing cluster equipped with eight high-performance GPUs. We used the Adam optimizer with a learning rate of 1e-4 for all models training. Model selection was based on the best performance on validation sets, and final evaluations were conducted on separate test sets. To ensure the reliability of our results, performance metrics—including averages and standard deviations—were reported across five random seeds.

## 3.3 EXPERIMENTAL RESULTS

**Anomaly Detection.** Table 2 compares AnyECG to state-of-the-art models in the anomaly detection task. AnyECG consistently outperforms other advanced models across all evaluation metrics. Specifically, the largest variant, AnyECG-XL, achieves the highest scores in accuracy, AUC-PR, AUROC, and Weighted F1 Score, demonstrating its strong ability to capture ECG signal characteristics. In contrast, traditional models like DENS-ECG (Peimankar & Puthusserypady, 2021) and ContraWR (Yang et al., 2021) show lower performance. DENS-ECG achieves moderate scores in accuracy and Weighted F1 Score, while ContraWR falls short in both metrics. Even the smaller versions of AnyECG, such as AnyECG-B and AnyECG-L, perform competitively and surpass most baseline models. This indicates that AnyECG maintains high performance across different scales without requiring extensive model parameters. Notably, the finetuned ECG-FM model (McKeen et al., 2024) performs at an intermediate to above-average level compared to the baseline. However, as a pre-trained model, its performance may still be hindered by substantial differences between the pre-training data and the downstream task dataset, which likely impedes its ability to fully converge.

**Arrhythmia Detection.** Table 3 presents a performance comparison between AnyECG and other leading models in arrhythmia detection. The results show that AnyECG, particularly the AnyECG-XL variant, consistently outperforms competing models across all metrics. This demonstrates its strong ability to handle arrhythmia detection effectively. In contrast, models like DENS-ECG (Peimankar & Puthusserypady, 2021) and ContraWR (Yang et al., 2021) exhibit lower per-

Table 2: Results Comparison with State-of-the-Art Models in Anomaly Detection

| Methods | Pretrain | Accuracy ↑ | AUC-PR ↑ | AUROC ↑ | Weighted F1 Score ↑ |
|---|---|---|---|---|---|
| DENS-ECG (Peimankar & Puthusserypady, 2021) | ✗ | 0.7928±0.0019 | 0.9319±0.0019 | 0.8488±0.0070 | 0.7928±0.0009 |
| ContraWR (Yang et al., 2021) | ✗ | 0.7551±0.0011 | 0.9374±0.0001 | 0.8153±0.0002 | 0.7611±0.0003 |
| XResNet1D (He et al., 2019) | ✗ | 0.7768±0.0115 | 0.9217±0.0045 | 0.7522±0.0121 | 0.7606±0.0093 |
| CNN-Transformer (Peh et al., 2022) | ✗ | 0.7401±0.0019 | 0.9340±0.0011 | 0.8074±0.0034 | 0.7444±0.0005 |
| RNN1D (Salloum & Kuo, 2017) | ✗ | 0.7992±0.0017 | 0.9284±0.0006 | 0.7868±0.0015 | 0.7838±0.0012 |
| FFCL (Li et al., 2022) | ✗ | 0.6709±0.0012 | 0.8682±0.0003 | 0.6423±0.0018 | 0.6746±0.0003 |
| Inception1D (Strodthoff et al., 2020) | ✗ | 0.8001±0.0029 | 0.9408±0.0004 | 0.8097±0.0015 | 0.7868±0.0018 |
| ST-Transformer (Song et al., 2021) | ✗ | 0.8070±0.0017 | 0.9471±0.0007 | 0.8406±0.0004 | 0.8048±0.0004 |
| ECG-FM (McKeen et al., 2024) | ✓ | 0.7788±0.0029 | 0.9036±0.0197 | 0.7693±0.0028 | 0.7321±0.0112 |
| AnyECG-B | ✓ | 0.8188±0.0025 | 0.9517± 0.0049 | 0.8502±0.0026 | 0.8863±0.0022 |
| AnyECG-L | ✓ | 0.8241±0.0043 | 0.9535±0.0030 | 0.8483±0.0025 | 0.8898±0.0026 |
| AnyECG-XL | ✓ | **0.8255±0.0035** | **0.9538±0.0012** | **0.8550±0.0016** | **0.8912±0.0033** |

formance. Notably, although ECG-FM (McKeen et al., 2024) employs pretraining, it achieves significantly lower accuracy. This underscores AnyECG's robustness, as its consistent performance across all metrics confirms its suitability for real-world arrhythmia detection.

Table 3: Results Comparison with State-of-the-Art Models in Arrhythmia Detection

| Methods | Pretrain | Accuracy ↑ | AUC-PR ↑ | Weighted F1 Score ↑ | Precision ↑ |
|---|---|---|---|---|---|
| DENS-ECG (Peimankar & Puthusserypady, 2021) | ✗ | 0.3202±0.0074 | 0.1514±0.0042 | 0.2669±0.0085 | 0.2866±0.0069 |
| ContraWR (Yang et al., 2021) | ✗ | 0.3075±0.0035 | 0.1359±0.0048 | 0.2802±0.0055 | 0.2794±0.0083 |
| XResNet1D (He et al., 2019) | ✗ | 0.1822±0.0058 | 0.1044±0.0011 | 0.1765±0.0031 | 0.1746±0.0124 |
| CNN-Transformer (Peh et al., 2022) | ✗ | 0.3284±0.0202 | 0.1417±0.0071 | 0.2685±0.0290 | 0.2641±0.0061 |
| RNN1D (Salloum & Kuo, 2017) | ✗ | 0.2511±0.0019 | 0.0911±0.0005 | 0.2164±0.0011 | 0.1986±0.0010 |
| FFCL (Li et al., 2022) | ✗ | 0.1823±0.0035 | 0.0832±0.0050 | 0.1770±0.0052 | 0.1736±0.0013 |
| Inception1D (Strodthoff et al., 2020) | ✗ | 0.2770±0.0031 | 0.1280±0.0006 | 0.2487±0.0031 | 0.2371±0.0021 |
| ST-Transformer (Song et al., 2021) | ✗ | 0.2011±0.0057 | 0.0941±0.0046 | 0.1996±0.0053 | 0.2018±0.0027 |
| ECG-FM (McKeen et al., 2024) | ✓ | 0.2212±0.0015 | 0.1037±0.0042 | 0.2285±0.0064 | 0.2386±0.0153 |
| AnyECG-B | ✓ | 0.3339±0.0029 | 0.1524±0.0069 | 0.2747±0.0046 | 0.3350±0.0052 |
| AnyECG-L | ✓ | 0.3358±0.0077 | 0.1542±0.0035 | 0.2636±0.0040 | 0.3339±0.0080 |
| AnyECG-XL | ✓ | **0.3449±0.0095** | **0.1635±0.0028** | **0.2833±0.0033** | **0.3449±0.0075** |

**Corrupted Lead Generation.** We evaluated AnyECG against CGAN (Mirza, 2014) and WGAN (Adler & Lunz, 2018) in generating corrupted ECG leads (see Table 4 and Figure 2). Using metrics like PSNR, SSIM, and MAE, AnyECG-L achieved the highest PSNR (32.7372 dB) and SSIM (0.8738), outperforming both CGAN and WGAN. Smaller models like AnyECG-L and AnyECG-B offer a better balance between capacity and generalization compared to AnyECG-XL. Due to limitations in its model architecture, ECG-FM (McKeen et al., 2024) could not be applied to this task. Although the AnyECG models did not achieve the lowest MAE, this may be because they prioritize capturing abstract rhythms and morphological patterns over minimizing pixel-level errors in detailed, noisy signals. This suggests that while AnyECG effectively captures the overall structure and rhythm of ECG signals, it is somewhat less precise in reproducing finer details. Figure 2 shows the ECG signals generated by WGAN, CGAN, and AnyECG. Both WGAN and CGAN can capture general morphology but fail to accurately reproduce certain rhythms, leading to unsuccessful signal generation in those cases. AnyECG leverages two stage pre-training to capture complex rhythmic features, resulting in morphology closer to the original signals. However, it lacks detailed feature extraction in finer wave bands, leading to poorer reconstruction in these regions and higher MAE. These observations suggest that while AnyECG excels in preserving overall rhythmic and morphological integrity, there is room for improvement in reconstructing fine-grained details.

**Ultra-Long ECG Recognition.**
Recognizing ultra-long ECG signals is challenging due to their extended duration, rhythm variability, and noise, which require models to be robust and generalizable. Traditional time series models often struggle with high computational complexity, memory constraints, and difficulty in capturing long-term dependencies. Therefore, We proposed a hierarchical modeling approach that adapts to ultra-long ECG data by employing a sliding window method.

Table 4: Results Comparison with State-of-the-Art Models in Corrupted Lead Generation.

| Methods | PSNR ↑ | SSIM ↑ | MAE ↓ |
|---|---|---|---|
| CGAN (Mirza, 2014) | 30.1762 | 0.8591 | **0.0142** |
| WGAN (Adler & Lunz, 2018) | 27.5074 | 0.7907 | 0.0199 |
| AnyECG-B | 32.5456 | 0.8634 | 0.0312 |
| AnyECG-L | **32.7372** | **0.8738** | 0.0296 |
| AnyECG-XL | 32.4276 | 0.8529 | 0.0376 |

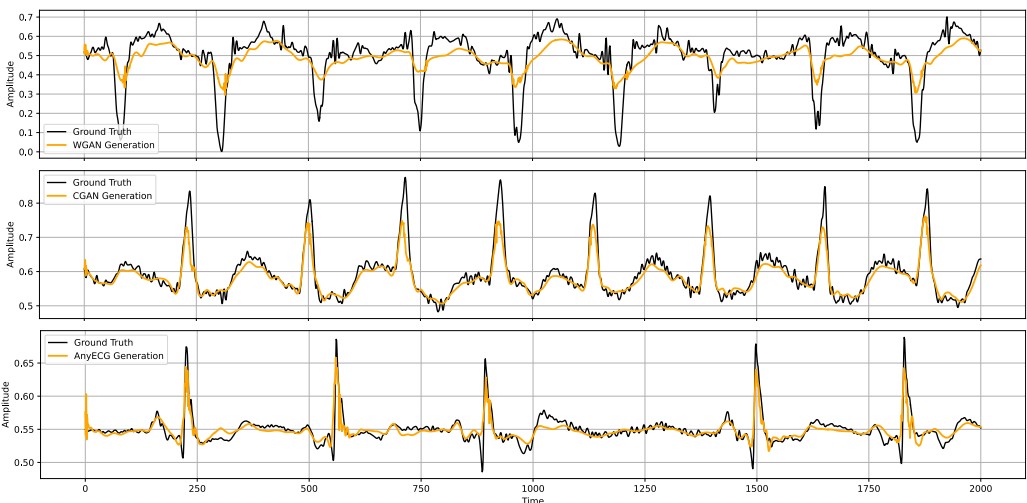

Figure 2: **Visualization of Corrupted Lead Generation** among WGAN (top), CGAN (middle), AnyECG (bottom).

As shown in Table 5, AnyECG, particularly the AnyECG-XL, achieves the highest scores across all evaluation metrics. This demonstrates its superior ability to capture complex patterns and maintain high accuracy when analyzing ultra-long ECG signals. Compared to state-of-the-art models like Inception1D (Strodthoff et al., 2020) and RNN1D (Salloum & Kuo, 2017), AnyECG-XL shows a clear advantage, especially in AUROC and AUC-PR. Even the smaller variants, AnyECG-B and AnyECG-L, outperform most baseline models, highlighting AnyECG's adaptability and scalability. The absence of results for the other pretrained ECG foundation model ECG-FM (McKeen et al., 2024) is due to its inability to handle ultra-long sequence data, making it unsuitable for this task. In contrast, AnyECG's consistent performance across all scales confirms its effectiveness in capturing key features of ultra-long ECG signals.

Table 5: Results Comparison with State-of-the-Art Models in Ultra-Long ECG Recognition

| Methods | Adaptation | Accuracy ↑ | AUC-PR ↑ | AUROC ↑ | Weighted F1 Score ↑ |
|---|---|---|---|---|---|
| DENS-ECG (Peimankar & Puthusserypady, 2021) | ✗ | 0.3202±0.0074 | 0.1514±0.0042 | 0.2669±0.0085 | 0.2866±0.0069 |
| ContraWR (Yang et al., 2021) | ✗ | 0.3075±0.0035 | 0.1359±0.0048 | 0.2802±0.0055 | 0.2794±0.0083 |
| XResNet1D (He et al., 2019) | ✗ | 0.6611±0.0812 | 0.6916±0.0797 | 0.6499±0.1353 | 0.6453±0.0922 |
| CNN-Transformer (Peh et al., 2022) | ✗ | 0.3284±0.0202 | 0.1417±0.0071 | 0.2685±0.0290 | 0.2641±0.0061 |
| RNN1D (Salloum & Kuo, 2017) | ✗ | 0.7444±0.0102 | 0.7724±0.0102 | 0.8679±0.0291 | 0.7386±0.0640 |
| FFCL (Li et al., 2022) | ✗ | 0.1823±0.0035 | 0.0832±0.0050 | 0.1770±0.0052 | 0.1736±0.0013 |
| Inception1D (Strodthoff et al., 2020) | ✗ | 0.5000±0.0017 | 0.5154±0.0492 | 0.3197±0.0573 | 0.3432±0.0038 |
| ST-Transformer (Song et al., 2021) | ✗ | 0.2011±0.0057 | 0.0941±0.0046 | 0.1996±0.0053 | 0.2018±0.0027 |
| AnyECG-B | ✓ | 0.6944±0.0016 | 0.7482±0.0025 | 0.6759±0.0056 | 0.5639±0.0124 |
| AnyECG-L | ✓ | 0.7777±0.0077 | 0.9075±0.0072 | 0.9104±0.0039 | 0.7500±0.0072 |
| AnyECG-XL | ✓ | **0.8055±0.0034** | **0.9088±0.0027** | **0.9104±0.0147** | **0.7741±0.0068** |

## 4 RELATED WORKS

**Heterogeneous ECG Signal Analysis and Classification.** The application of deep learning techniques has significantly advanced the analysis and classification of ECG signals. However, the heterogeneity of ECG data poses a major challenge for model generalization; models trained on one dataset often do not perform well on others. Consequently, researchers have focused on designing specialized models tailored to specific datasets, employing architectures such as convolutional neural networks (CNNs) (Prathipati & Malyavantham, 2023; Kucukseymen et al., 2022), recurrent neural networks (RNNs) (Kumar et al., 2023; Din et al., 2024), and transformer-based models (Shah et al., 2024; Ji et al., 2024). While these efforts have led to incremental performance improvements (Srivastava et al., 2023; Jasvitha et al., 2024; Ribeiro et al., 2020; Gao et al., 2021), the gains are often not statistically significant due to the limited size and scope of the datasets used. The absence of a unified model capable of handling the diverse nature of ECG data underscores the need for new approaches that can provide more substantial and broadly applicable performance improvements.

**Self-supervised ECG Representation Learning.** Self-supervised learning has emerged as a promising approach for extracting representations from unlabeled ECG signals, enabling the use of large amounts of raw data without manual annotations. Methods such as signal reconstruction, contrastive learning, and masked signal modeling have been explored (Yun et al., 2024; Wu et al., 2024; Li et al., 2024). However, existing self-supervised learning methods often struggle to generalize across heterogeneous ECG datasets, especially when faced with varying lead configurations and noise levels. For example, contrastive methods (Kiyasseh et al., 2021; Wang et al., 2023) encourage similar representations for compatible signal segments but do not adequately account for variability introduced by different lead setups. Moreover, the low SNR inherent in ECG data can cause models to focus on reconstructing noisy or redundant signal components due to high correlations among leads, rather than capturing critical physiological information. Models like contrastive predictive coding (CPC) (Mehari & Strodthoff, 2022) and masked autoencoders (Zhang et al., 2022; Na et al., 2024) often inadvertently emphasize less relevant features, diminishing their effectiveness in capturing essential signal characteristics. This focus on less informative aspects can limit the models' ability to extract meaningful representations that transfer effectively to unseen data or datasets with different characteristics.

## 5 Discussion

**Social Impacts.** ECG is one of the most commonly used diagnostic tools in healthcare, with over 100 million ECG reports obtained annually in the United States alone (Tison et al., 2019). Despite its widespread use, unlike other biomedical signals such as electroencephalograms (EEG) (Yang et al., 2024; Jiang et al., 2024), there is a scarcity of foundation models specifically designed for ECG data. This limitation hampers the potential for advanced analysis and interpretation of ECG signals on a large scale. In this work, we propose AnyECG, the largest ECG foundation model family to date. Compared to prior works (McKeen et al., 2024; Song et al., 2024; Fu et al., 2024), AnyECG adapts to diverse downstream tasks and achieves significantly better performance. By providing a robust and generalizable model for ECG data, AnyECG has the potential to greatly enhance diagnostic accuracy, facilitate early detection of cardiovascular diseases, and improve patient outcomes on a broad scale.

**Limitations.** Although we pre-trained AnyECG using a large amount of data across seven datasets, there remains a significant gap between AnyECG and current foundation models like LLMs in the general domain. This gap is primarily due to the difficulty in obtaining extensive healthcare data. Additionally, the model size of AnyECG-XL (1.7B parameters) is considerably smaller than that of foundation models in natural language processing and computer vision fields. Despite these limitations, it is important to highlight that training a large-scale ECG foundation model with a two-stage self-supervised learning approach and more data does yield appreciable performance gains compared to existing methods developed for specific downstream tasks, even if it may be computationally costly. Exploring the trade-off between employing larger AnyECG models and enhancing downstream task performance will be a focus of our future work.

## 6 Conclusion

In this paper, we proposed AnyECG, a foundation model family that learns universal embeddings through a two-stage self-supervised pre-training on seven diverse ECG datasets. AnyECG effectively handles the heterogeneity of ECG data through the design of a novel ECG Tokenizer, which includes a rhythm codebook and a multi-view synergistic decoder to learn representations from different proxy tasks. Additionally, the masked modeling in the second-stage pre-training plays a crucial role in enabling effective representation learning of both temporal and lead features of ECG signals. We validated various sizes of AnyECG models on multiple downstream tasks, including anomaly detection, arrhythmia detection, ECG lead generation, and ultra-long ECG signal recognition. Our experiments demonstrate that AnyECG outperforms all state-of-the-art methods in their respective fields, highlighting its effectiveness and versatility in ECG signal analysis.

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

# 7 APPENDIX

## CONTENTS

## 7.1 ECG DATASETS

To evaluate the performance of AnyECG and baseline models, we utilized a comprehensive set of ECG datasets that cover a wide spectrum of cardiac conditions, patient demographics, and recording scenarios, ensuring robust testing across diverse settings. The datasets include:

**CPSC and CPSC-Extra Databases (Liu et al., 2018):** These consist of 12-lead ECG recordings ranging from 6 to 60 seconds in duration, sampled at 500 Hz, and include a balanced mix of male and female subjects.

**INCART Database (Tihonenko et al., 2008):** This database provides 74 annotated recordings extracted from 32 Holter records, each 30 minutes long and sampled at 257 Hz, offering high-resolution data ideal for arrhythmia classification.

**PTB (Bousseljot et al., 1995) and PTB-XL Databases (Wagner et al., 2020):** The PTB Diagnostic ECG Database includes 516 recordings sampled at 1000 Hz, while PTB-XL contains 21,837 12-lead ECGs sampled at 500 Hz, each 10 seconds long, encompassing a wide range of cardiac pathologies.

**Georgia 12-lead ECG Challenge (G12EC) Database:** Comprising 10,344 recordings from the Southeastern United States, sampled at 500 Hz, this dataset adds demographic diversity to our evaluation.

**Undisclosed Database:** This dataset contributes an additional 10,000 ECG recordings, providing a geographically distinct test set to further validate the model's performance without data leakage.

By employing this diverse collection of datasets, we thoroughly assess AnyECG's ability to generalize across different patient populations, signal qualities, and clinical conditions.

## 7.2 EVALUATION METRICS

We conducted four distinct experiments, each utilizing a specific set of evaluation metrics tailored to the task:

1. **Anomaly Detection:** Evaluated using **Accuracy**, **AUC-PR** (Area Under the Precision-Recall Curve), **AUROC** (Area Under the Receiver Operating Characteristic Curve), and **Weighted F1 Score**. These metrics assess the model's ability to correctly identify anomalies and handle class imbalances effectively.

2. **Arrhythmia Detection:** Assessed with **Accuracy**, **AUC-PR**, **Weighted F1 Score**, and **Precision**. This combination of metrics evaluates the model's performance in detecting various types of arrhythmias, emphasizing both overall accuracy and the precision of positive predictions.

3. **Corrupted Lead Generation:** Measured using **PSNR** (Peak Signal-to-Noise Ratio), **SSIM** (Structural Similarity Index), and **MAE** (Mean Absolute Error). These metrics quantify the quality of the generated ECG signals by comparing the reconstructed signals to the original ones, focusing on signal fidelity and structural similarity.

4. **Ultra-Long ECG Recognition:** Evaluated with **PSNR**, **SSIM**, and **MAE**, similar to the corrupted lead generation task. These metrics ensure that the model maintains high-quality signal reconstruction and accurate recognition over extended ECG recordings.

The definitions of the evaluation metrics used across these experiments are as follows: **Accuracy**: The proportion of correctly predicted instances out of all instances, indicating the overall effectiveness of the model. **Precision**: The ratio of true positive predictions to the total number of positive predictions, reflecting the model's ability to avoid false positives. **AUC-PR** (Area Under the Precision-Recall Curve): Measures the trade-off between precision and recall for different threshold settings, particularly useful for imbalanced datasets. **AUROC** (Area Under the Receiver Operating Characteristic Curve): Represents the model's ability to distinguish between classes across all classification thresholds. **Weighted F1 Score**: The harmonic mean of precision and recall, weighted by the number of true instances for each class, providing a balanced evaluation of the model's performance. **PSNR** (Peak Signal-to-Noise Ratio): Indicates the quality of signal reconstruction by comparing the maximum possible signal power to the power of reconstruction noise, with higher values signifying better quality. **SSIM** (Structural Similarity Index): Assesses the similarity between two signals in terms of luminance, contrast, and structure, with values closer to 1 indicating higher similarity. **MAE** (Mean Absolute Error): Represents the average absolute difference between predicted and actual values, serving as a measure of prediction accuracy. By employing these tailored metrics across different experiments, we ensure a comprehensive evaluation of our model's performance in various aspects of ECG signal processing and classification.

## 7.3 PRE-TRAINING PHASE ABLATION STUDY

To evaluate the contribution of each component in our pre-training strategy, we conducted an ablation study focusing on the pre-training phases. Specifically, we analyzed the effects of pre-training the ECG Tokenizer and the full AnyECG foundation model on anomaly detection performance. Table 6 and Figure 3 presents the results of this study. The first configuration is AnyECG-B without ECG Tokenizer pre-training. The second configuration includes a pre-trained ECG Tokenizer but skips pre-training the AnyECG foundation model. The final configuration involves full pre-training of both the ECG Tokenizer and the AnyECG. The results show that pre-training the ECG Tokenizer leads to noticeable improvements over the baseline. This indicates that a pre-trained ECG Tokenizer enhances the model's ability to capture meaningful representations of the ECG signals.

When the full AnyECG foundation model is also pre-trained, we observe a significant performance boost across all metrics. These gains underscore the importance of comprehensive pre-training in enhancing the model's anomaly detection capabilities. The fact that full pre-training yields the best results confirms that both components—the ECG Tokenizer and the AnyECG—contribute positively to the overall performance. Pre-training the AnyECG foundation model allows it to learn generalizable features that are beneficial for downstream tasks, while the pre-trained ECG Tokenizer ensures effective encoding of the input signals.

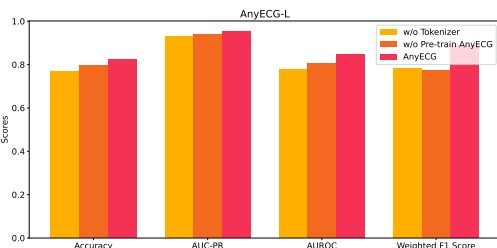 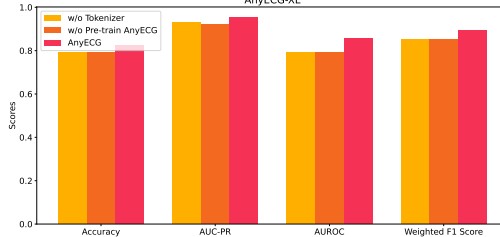

Figure 3: Ablation study of Pre-training Phase in Anomaly Detection with AnyECG-L and AnyECG-XL

Table 6: Ablation study of Pre-training Phase in Anomaly Detection

| Methods | Accuracy ↑ | AUC-PR ↑ | AUROC ↑ | Weighted F1 Score ↑ |
|---|---|---|---|---|
| AnyECG-B w/o ECG Tokenizer | 0.7623 | 0.9243 | 0.7729 | 0.7512 |
| AnyECG-B w/o Pre-train AnyECG | 0.7810 | 0.9358 | 0.8232 | 0.7826 |
| AnyECG-B | **0.8188** | **0.9517** | **0.8502** | **0.8863** |

### 7.4 LOSS FUNCTION ABLATION STUDY

To assess the effectiveness of our loss function design, we conducted an ablation study by systematically removing each component of the loss function. This allowed us to evaluate how each term contributes to the model's ability to capture meaningful features from ECG signals. Table 7 presents the results of this study. The "Full Loss" configuration, which includes all components of our proposed loss function, serves as the baseline for optimal performance. When we individually removed each loss component, we observed a decrease in performance across all evaluation metrics. Omitting the Morphology Loss resulted in a noticeable decline, indicating its significant role in helping the model capture the morphological characteristics of ECG signals, which are crucial for accurate anomaly detection. Excluding the Frequency Loss also led to reduced performance, suggesting that capturing frequency domain information is important for understanding underlying patterns in ECG signals. Removing the Demography Loss caused a performance drop as well, though to a lesser extent compared to the Morphology and Frequency losses. This highlights that incorporating demographic information refines the model's predictions by accounting for variations in ECG patterns across different demographic groups. The most significant decrease in performance was observed when the Codebook Loss was removed. This component is essential for encouraging diversity and utilization of codebook entries in the vector quantization process, playing a critical role in the model's ability to represent ECG signals effectively. Lastly, removing the Commitment Loss also led to a decline in performance, though the impact was less severe than omitting the Codebook Loss. The Commitment Loss ensures consistency in the representation of similar inputs by encouraging the encoder to commit to specific codebook entries. The combination of Morphology, Frequency, Demography, Codebook, and Commitment losses enables the model to capture comprehensive features of ECG signals, leading to improved anomaly detection capabilities. These results validate the design of our loss function and underscore the importance of each term in capturing different aspects of the ECG data.

Table 7: Ablation Study on Loss Function Components in Anomaly Detection

| Loss Configuration | Accuracy ↑ | AUC-PR ↑ | AUROC ↑ | Weighted F1 Score ↑ |
|---|---|---|---|---|
| Full Loss | **0.8188** | **0.9517** | **0.8502** | **0.8863** |
| w/o Morphology Loss | 0.8059 | 0.9373 | 0.8381 | 0.8754 |
| w/o Frequency Loss | 0.8125 | 0.9412 | 0.8475 | 0.8621 |
| w/o Demography Loss | 0.8134 | 0.9487 | 0.8445 | 0.8801 |
| w/o Codebook Loss | 0.7522 | 0.8950 | 0.7900 | 0.8150 |
| w/o Commitment Loss | 0.7855 | 0.9275 | 0.8225 | 0.8575 |

## 7.5 NOTATIONS

**Data and Indices**

| | |
|---|---|
| $X \in \mathbb{R}^{L \times T}$ | Multi-channel ECG signals |
| $L$ | Number of ECG leads |
| $T$ | Total time steps |
| $s$ | Patch size (time steps) |
| $P$ | Number of patches per lead |
| $x_{j,k} \in \mathbb{R}^s$ | Patch from lead $j$, index $k$ |
| $N = L \times P$ | Total number of patches |
| $j = 1, 2, \ldots, L$ | Lead index |
| $k = 1, 2, \ldots, P$ | Patch index |

**Embeddings and Positional Encodings**

| | |
|---|---|
| $d$ | Embedding dimension |
| $h'_{j,k} \in \mathbb{R}^d$ | Patch embedding |
| $\tau_k \in \mathbb{R}^d$ | Temporal positional encoding |
| $\sigma_j \in \mathbb{R}^d$ | lead positional encoding |
| $h_{j,k} = h'_{j,k} + \tau_k + \sigma_j$ | Augmented embedding |

**Neural Tokenizer and Codebook**

| | |
|---|---|
| $V \in \mathbb{R}^{K \times d}$ | Codebook of codewords |
| $i^*$ | Codeword index |
| $z_{j,k} = i^*$ | Discrete token |
| $K$ | Number of codewords |

**Attention Mechanism and Transformer Components**

| | |
|---|---|
| $H \in \mathbb{R}^{N \times d}$ | Embedding matrix |
| $Q, K, V$ | Query, key, value matrices |
| $W_Q, \ W_K, \ W_V \ \in \ \mathbb{R}^{d \times d_{\text{model}}}$ | Projection matrices |
| $\text{LayerNorm}(\cdot)$ | Layer normalization |
| $M \in \mathbb{R}^{N \times N}$ | Attention mask |
| $\mathcal{A}(i)$ | Attention set for patch $i$ |
| $d_{\text{head}}$ | Head dimension |
| $d_{\text{model}}$ | Model dimension |

**Decoders and Reconstruction**

| | |
|---|---|
| $o^m_{j,k}$ | Morphology Decoder output |
| $c^{(l)}_A[n]$ | Approximation coefficients |
| $c^{(l)}_D[n]$ | Detail coefficients |
| $c^{(l)\,\text{norm}}_A, c^{(l)\,\text{norm}}_D$ | Normalized coefficients |
| $\hat{c}^{(l)}_A, \hat{c}^{(l)}_D$ | Predicted coefficients |
| $g[\cdot], h[\cdot]$ | Filter coefficients |
| $a \in \mathbb{R}^{d_a}$ | Demography vector |
| $o^a$ | Predicted demography |

| | |
|---|---|
| $\hat{x}_{j,k}$ | Reconstructed patch |
| $f$ | Decoder function |

**Loss Functions**

| | |
|---|---|
| $\mathcal{L}_{\text{morphology}}$ | Morphology loss |
| $\mathcal{L}_{\text{freq}}$ | Frequency loss |
| $\mathcal{L}_{\text{demography}}$ | Demography loss |
| $\mathcal{L}_{\text{codebook}}$ | Codebook loss |
| $\mathcal{L}_{\text{commitment}}$ | Commitment loss |
| $\mathcal{L}_T$ | Total tokenizer loss |
| $\mathcal{L}_{\text{mask}}$ | Masked modeling loss |

**Masking and Autoregression**

| | |
|---|---|
| $m_{j,k} \in \{0, 1\}$ | Mask indicator |
| $h_M \in \mathbb{R}^d$ | Mask token |
| $\tilde{h}_{j,k}$ | Masked embedding |
| $\bar{h}_{j,k} = \tilde{h}_{j,k} + \tau_k + \sigma_j$ | Augmented masked embedding |
| $\tilde{h}'_{j,k}$ | Contextualized representation |
| $p(v_i \mid \tilde{H})$ | Probability distribution |

**Other Parameters and Hyperparameters**

| | |
|---|---|
| $L_w$ | Decomposition levels |
| $d_a$ | Demography vector dimension |
| $n$ | Coefficient index |
| $r$ | Masking ratio |

