# OpenReview forum: "AnyECG: Foundational Models for Electrocardiogram Analysis"
_ICLR.cc/2025/Conference — Submitted to ICLR 2025_

### Official Review · Reviewer_7oXo · 2024-11-01

**Soundness:** 2
**Presentation:** 2
**Contribution:** 2
**Rating:** 3
**Confidence:** 5

**Summary:**

This paper aims to design a general ECG foundation model capable of addressing the heterogeneity of ECG tasks, low signal-to-noise ratio (SNR), demographic shifts, and identifying implicit rhythm-event associations in real-world scenarios. By proposing a standardized tokenizer approach for any ECG data and a mask-based ECG pretraining method, the authors seek to enable adaptive learning of rhythm attributes. The model achieves promising results across various downstream tasks.

**Strengths:**

1. The authors thoroughly consider the complexities in real-world ECG tasks, proposing a codebook-driven standardized ECG encoding tokenizer that leverages prior knowledge to facilitate learning of cross-lead and temporal information.

2. The paper introduces morphology and frequency decoders to jointly guide the model’s reinforced understanding of the time and frequency domains. Additionally, a demographic decoder is proposed to address cross-population heterogeneity, which is relatively rare in ECG studies.

**Weaknesses:**

1. The authors appear to lack foundational understanding of ECG, leading to a motivation and set of four challenges that do not align with the actual difficulties in ECG tasks. Their data processing and methods rely heavily on conventional NLP approaches, without adequately considering their applicability to ECG tasks. For example, padding 10-second, single- or three-lead ECG data to 12 leads over 60 seconds substantially increases data volume and may lead to erroneous lead-dropout interpretations. Furthermore, the direct conversion of patches from different times and channels into a single sequence risks confusing temporal and channel correlations post-transformer processing, which contradicts realistic ECG analysis logic.

2. The morphology decoder faces similar issues: masking the temporal signal alters the fragment’s attributes, potentially shifting to other ECG abnormalities, thereby hindering accurate reconstruction of temporal signal features.

3. The experimental design does not sufficiently demonstrate the model’s effectiveness in addressing real-world ECG challenges. The results focus excessively on technical validations, lacking a clear analysis of the model’s practical impact on ECG tasks. From the results, it is unclear whether this model qualifies as a foundation model.

**Questions:**

What is the basis for selecting a filter range of 0.1 – 75 Hz?

In lines 139-141, what distinguishes w from s? The definition of s is missing upon its first appearance. P is defined as patch number, w as patch size, yet line 343 defines P=300, which seems to imply w=300.

For the ECG tokenizer, what is the positional encoding method for /tau_k and /sigma_j? Does encoding along the channel dimension hold any significance? If so, what does it represent?

How is the vector-quantized rhythm codebook (line 180) implemented? What does it entail, and how is it proven that a vector can replace the original ECG signal encoding?

What does anomaly refer to in line 361, and which database was used for this experiment?

Which databases were used for the arrhythmia detection task? How should the metrics, all under 0.35, be interpreted? Is this model meaningful for ECG tasks? For example, the AUC achieved on the PTBXL database reached 0.9 in CinC/PhysioNet 2021 results.

In Figure 2, the horizontal axis should be labeled as Time (s) rather than points, and the amplitude on the vertical axis should confirm if its unit is mV. Additionally, the generated signal significantly deviates from actual ECG characteristics, lacking key wave features. How do the authors justify that the generated signal is usable for downstream ECG tasks?

What is the length of the ultra-long ECG? Can it handle 1-hour ECG data or longer? How is the dimension M managed within the transformer for this data length?

The database dimensions should be clearly described. The authors claim to train a foundation model but did not include wearable signals (single-lead). How can this model be substantiated as an ECG foundation model?

---

### Official Review · Reviewer_cHqi · 2024-11-02

**Soundness:** 2
**Presentation:** 3
**Contribution:** 2
**Rating:** 3
**Confidence:** 5

**Summary:**

This manuscript presents AnyECG, a novel framework for ECG representation learning. This framework consists of a Transformer-based tokenizer, a Transformer-based encoder, and a self-supervised learning framework. The self-supervised learning framework is based on masked learning. The tokenizer takes patches of ECG signals of fixed length and converts them into discretized tokens. The authors conducted experiments on several ECG datasets to evaluate the performance of AnyECG.

**Strengths:**

1. The authors made a good try to apply masked learning to ECG representation learning. Previous works mainly focus on contrastive learning.

2. The figure of the overall architecture of AnyECG is quite nice and informative.

**Weaknesses:**

1. The authors did not compare AnyECG with contrastive learning-based methods, which are the most related works to AnyECG.

2. It is hard to say that the datasets used in the experiments are representative enough. The datasets used in this work are only a part of the CinC2021 challenge dataset (ref. https://physionet.org/content/challenge-2021/1.0.3/, the Chapman-Shaoxing and Ningbo Database and The University of Michigan (UMich) Database are not used in this work), which are all standard 12-lead ECG datasets. The authors should use more comprehensive datasets to make the name "AnyECG" more convincing, for example, ambulatory ECG datasets collected from wearable devices, or datasets of much larger size (https://zenodo.org/records/4916206), etc.

3. The terminologies used in this work are messy. For example, on page 3, in the paragraph just beneath Figure 1, "w" and "s" are used interchangeably to represent the length of the patch. The authors should use a consistent notation.

**Questions:**

1. The authors stated that they used an "Undisclosed Database" in the experiments. However, according to its description in this manuscript, this database is among the hidden test data of the CinC2021 challenge dataset, which was claimed never to be made public. I'm curious about how the authors obtained this dataset, or if some of the organizers of the CinC2021 challenge dataset are among the authors of this work.

2. One of the downstream tasks for training the tokenizer is morphology reconstruction, which has equal weights across the whole signal. However, the waveforms (namely the P wave, QRS complex, and T wave) in ECG signals have much more importance than other parts of the signal. Should one assign different weights to different parts of the signal for better performance? Another question is that small shifts along the time axis near the waveforms, especially the QRS complex, may lead to significant increases in the total loss. How does the tokenizer handle this issue?

---

### Official Review · Reviewer_1p2A · 2024-11-04

**Soundness:** 2
**Presentation:** 2
**Contribution:** 2
**Rating:** 3
**Confidence:** 5

**Summary:**

This paper introduces a foundational model for ECG analysis designed to handle the ECG data collected from various devices and scenarios. Recognizing challenges in ECG representation learning—such as data heterogeneity, low signal-to-noise ratio (SNR), demographic variability, and complex rhythm-event associations—the authors propose a method with two main components:

- ECG Tokenizer: This component encodes each ECG fragment into discrete tokens, using a Rhythm Codebook to capture essential morphological, frequency, and demographic information. This approach improves the model's ability to manage noisy signals and demographic shifts while preserving clinically relevant features.

- Hierarchical Model for Ultra-Long ECGs: AnyECG employs a hierarchical modeling approach to support ultra-long ECG sequences, using a sliding window technique to capture extended cardiac rhythms and events effectively.

The model is pre-trained on diverse datasets and evaluated on multiple downstream tasks, including anomaly detection, arrhythmia classification, corrupted lead generation, and ultra-long ECG recognition.

**Strengths:**

The paper appropriately identifies the issue of heterogeneity in ECG data and provides a well-founded problem definition to address it.

**Weaknesses:**

The authors propose a method to address heterogeneity in ECG data, but the dataset used for downstream applications does not sufficiently reflect this heterogeneity. The model should be tested not only on standard 12-lead ECGs but also on ECGs obtained from diverse measurement devices, including treadmill tests, Holter monitors, patches, and consumer-driven mobile ECG devices.

Using 1-second tokens could be limiting, as electrical signals in ECGs can display features in the range of tens to hundreds of milliseconds. Therefore, it is necessary to verify whether the model can effectively classify critical cardiac abnormalities, such as acute myocardial infarction or acute QT prolongation, which occur on a sub-second timescale in addition to arrhythmias.

According to the paper, the anomaly detection task is described as identifying irregularities in ECGs; however, these irregularities are characteristic of arrhythmias, making it challenging to consider this as a separate task.

**Questions:**

In the case of arrhythmia detection tasks using deep learning, accuracy is typically very high; however, Table 3 does not reflect this. Additional explanation is needed regarding which specific arrhythmias this task is diagnosing and how labeling was structured for each benchmark dataset.

The evaluation metrics used in this paper focus on identifying whether the input ECG is anomalous or indicative of arrhythmia. However, for long-term ECGs, it is essential that the model also pinpoints the location of abnormal ECG rhythms. Including metrics that can evaluate this aspect of performance would be beneficial.

---

### Official Review · Reviewer_B1H3 · 2024-11-04

**Soundness:** 3
**Presentation:** 2
**Contribution:** 1
**Rating:** 3
**Confidence:** 5

**Summary:**

This paper divides the pre-training stage into two steps: ECG tokenizer pre-training and encoder pre-training. This paper trains the tokenizer to be ECG-specific. Additionally, it aims to enhance robustness by utilizing a Rhythm codebook.

**Strengths:**

By training the tokenizer to be ECG-specific, rather than relying on simple linear layers commonly used in previous studies, the paper seeks to enhance the model's relevance to ECG data. Furthermore, it aims to enhance robustness through the utilization of a Rhythm codebook.

**Weaknesses:**

1. **Lack of Experiments Addressing Key Challenges**:
   - The paper does not provide sufficient experiments to verify whether the mentioned challenges are effectively resolved. The ablation studies are inadequate, with none in the main text and only limited ones in the appendix.

2. **Insufficient Baseline Comparisons**:
   - Despite aiming to position itself as a foundation model, the paper fails to compare its approach with other foundational or self-supervised learning (SSL) methods, which is a critical oversight.

3. **Incomplete Downstream Task Descriptions**:
   - The explanation of downstream tasks is lacking. It is unclear if the same datasets used for pre-training were also used for downstream tasks. Specific details on the datasets, labels, and tasks (e.g., anomaly detection) are missing.
   - Not all the datasets have the labels for anomlay. What is the downstream dataset for anomaly detection?
   - There is no clear explanation of how corrupted lead generation was conducted or the settings involved (e.g., generating precordial leads from limb leads).
   - Did you use the morphology decoder for the generation of corrupted leads?
   - The methodology and setting for Ultra-Long ECG Recognition are not described, and the meaning of "adaptation" in Table 5 is not explained in the text.
   - Moreover, in Section 7.2, the evaluation metrics for Ultra-Long ECG Recognition are listed as PSNR, SSIM, and MAE, but this does not align with what is shown in Table 5. Which one is correct?

4. **Presentation Issues**:
   - The notation for patch size is inconsistent, using both $w$ and $s$ interchangeably.
   - Figure 1 (bottom-left) mentions using N transformer blocks, but this contradicts the text, where N refers to the total number of patches.

**Questions:**

1. **Proxy Task Explanation**:
   - In the demographic task, it is stated that the model predicts age, gender, race, and, etc. How does the model handle datasets that lack one or more of these attributes? Please provide a detailed explanation.

2. **Corrupted Lead Generation**:
   - Why is the Mean Absolute Error (MAE) high in the corrupted lead generation task? Please explain in detail.
   - Why were PSNR and SSIM chosen as metrics for this task?
   -  In Figure 2, the generation results are shown for different ground truth (GT) signals. It would be helpful to see the results for the same GT signal for comparison. Could you provide this?

3. **Challenge Clarification**:
   - What exactly is meant by "Implicit rhythm-event association" mentioned as a challenge? Please clarify.

4. **Comparison with Baselines**:
   - Did you measure the training time and inference time compared to other baseline methods? If so, what were the results?

---

### Meta-Review · Area_Chair_Uzyi · 2024-12-19

**Metareview:**

The paper proposes a foundation model for ECG analysis using a two-stage training approach. The paper develops a custom tokenizer to convert continuous ECG signals into discrete tokens. The core idea is to use a Rhythm Codebook to encode morphological, frequency, and demographic information. The encoder model is trained on to learn representations of very long ECG segments (ultra-long ECGs), using a hierarchical modeling framework to capture extended cardiac rhythms and events. The model is evaluated on several downstream tasks, including anomaly detection, arrhythmia classification, corrupted lead generation, and ultra-long ECG recognition. While I think the Rhythm Codebook is an interesting idea that I encourage the authors to develop further, the reviewers found this version of the work was inadequate. The comments highlighted the need for more experimental validation, the need for more motivation for metrics and expanding the empirical evaluation to also include an expanded set of 12-lead ECGs.

**Additional Comments On Reviewer Discussion:**

There was no response to reviews provided by the authors.

---

### Decision · Program_Chairs · 2025-01-22

Reject